# Comparative Analysis on the Effect of Sarcopenia in Patients with Knee Osteoarthritis before and after Total Knee Arthroplasty

**DOI:** 10.3390/diseases11010036

**Published:** 2023-02-22

**Authors:** Chrysanthi Liliana Tzartza, Nikolaos Karapalis, Gavriela Voulgaridou, Christiana Zidrou, Anastasios Beletsiotis, Ioanna P. Chatziprodromidou, Constantinos Giaginis, Sousana K. Papadopoulou

**Affiliations:** 1Department of Nutritional Sciences and Dietetics, International Hellenic University, 57400 Thessaloniki, Greece; 22nd Orthopaedic Department, G. Papageorgiou General Hospital, 54453 Thessaloniki, Greece; 3Department of Public Health, Medical School, University of Patras, 26504 Patra, Greece; 4Department of Food Science and Nutrition, School of Environment, University of Aegean, 81400 Myrina, Lemnos, Greece

**Keywords:** sarcopenia, osteoarthritis, total knee arthroplasty, muscle strength, muscle mass

## Abstract

Introduction: Primary sarcopenia is an age-related disease that occurs mainly in older adults, while its possibility of appearance increases with age. Secondary sarcopenia is related to the presence of a disease. At times, studies have implied a connection between various diseases and the appearance of sarcopenia. Due to pain, patients with knee osteoarthritis limit their everyday activities, leading to a decrease in muscle mass and physical function. Purpose: This study aimed to investigate the impact of the coexistence of sarcopenia and osteoarthritis on patients’ rehabilitation and symptoms, such as pain, after total knee arthroplasty, compared with patients with osteoarthritis without sarcopenia. Methodology: This cross-sectional study material consisted of 20 patients with osteoarthritis, who were hospitalized at Papageorgiou Hospital of Thessaloniki for total knee arthroplasty from November 2021 to April 2022. The patients were evaluated for sarcopenia according to the FNIH criteria. The two groups were asked to complete the KOOS score questionnaire in order to evaluate the condition of their knee in two phases, before surgery and 3 months after surgery. Results: The two groups, 5 sarcopenic patients and 15 non-sarcopenic, did not show a statistically significant difference in muscle strength measurements. However, the lean mass indices, ALM (15.18 ± 3.98 versus 19.96 ± 3.65, respectively; *p* = 0.023) and ALM/height^2^ (5.53 ± 1.40 versus 6.98 ± 0.75, respectively; *p* = 0.007) had significant differences, since the sarcopenic group showed a reduced lean mass, especially in patients with a comorbidity of cancer. Sarcopenic patients showed a smaller increase in KOOS score compared to non-sarcopenic patients before (0.38 ± 0.09 vs. 0.35 ± 0.09, respectively; *p* = 0.312) and after surgery (0.54 ± 0.08 vs. 0.59 ± 0.10, respectively; *p* = 0.909), but without a statistically significant difference. The score increased for both groups, with the time factor playing a greater role than the group. Conclusions: Both the sarcopenic group and the control group did not show significant differences in their scores for the assessment of the affected limb in any of the two phases while completing the questionnaire. However, there was an improvement in their osteoarthritis symptoms before and after arthroplasty in both groups. Further research with a larger sample and longer recovery time is needed to draw more accurate conclusions and confirm the present results.

## 1. Introduction

Sarcopenia constitutes a pathological condition accompanied by a gradual decrease in muscle mass, as well as loss of muscle function and strength [1]. The main factors or situations that lead to the degradation of muscle tissue are still under research [2]. Although there is no sufficient explanation, there are many possible proving factors, such as satellite cells, inflammation, fibroblast growth factors (FGF), hormonal factors, autophagy, myosteatosis, reactive oxygen species (ROS), p38 mitogen-activated protein kinases (p38MAPK), and p16Iu4a [2]. The multifactorial pathophysiological pathways of sarcopenia make it difficult to find out which specific mechanism causes this syndrome. In addition, different lifestyle factors augment even more the difficulty of interpreting sarcopenia. Thus, the study of lifestyle factors needs extended research. Lifestyle factors such as physical activity, nutrition/diet, sleep duration and quality, unhealthy habits or substances (nicotine and alcohol), etc., affect the expression of sarcopenia, reversing or exacerbating the symptoms. Patients’ conditions worsen with advanced age, as the prevalence of the disease increases sharply after the age of 60 years [3]. The prevalence of sarcopenia in different regions ranges around 10% for people over 60 years regardless of gender, with lower rates for Asians compared to the rest of world [4]. A recent meta-analysis showed small differences between men and women. The prevalence in men is slightly higher (11%) compared to women (9%) and much higher in elderly nursing home residents and patients (31–51% and 23–24% for men and women, respectively) [5]. As age increases, various changes occur; physical activity decreases, dietary habits change, the gut microbiome changes, and the hormonal balance in the body is disturbed, resulting in a change in body composition and a decrease in muscle mass and strength [6]. Sarcopenia coexists with several other diseases and worsens the condition of these patients, increasing hospitalization time and mortality rates [7,8]. Diabetes mellitus is associated with adverse changes in body composition due to metabolic disturbances that can cause inflammation and oxidative stress [9]. Furthermore, the prevalence of sarcopenia appears to be quite elevated in patients with cancer, exceeding 50% in elderly cancer patients [10]. Sarcopenic cancer patients have increased rates of complications and mortality compared to non-sarcopenic patients [11]. There is a two-way relationship between sarcopenia and cancer, since patients with cancer are at higher risk of developing sarcopenia than the general population due to the increased inflammatory response of the body and invasive treatments that patients undergo [12]. Early diagnosis challenges the health care system because of the different parameters and criteria that exist. Hence, the prevention of sarcopenia would be life-saving in order to avoid the burden that it imposes on both the patient and the health care system [13].

Osteoarthritis is also a very common disease that occurs in older adults and is the most frequent reason of physical inactivity. Due to low physical levels, patients with osteoarthritis have a more than ~20% higher mortality rate compared to patients without osteoarthritis [14]. Osteoarthritis is a disease characterized by a reduction in the articular cartilage surface, as well as damage to the subchondral bone and synovial membrane. Inflammatory factors lead to an imbalance between anabolic and catabolic activities, resulting in a reduction in chondrocytes [15]. Pro-inflammatory cytokines, such as interleukin 6 (IL-6), interleukin 8 (IL-8), monokine induced by gamma (MIG), macrophage inflammatory protein-1β (MIP-1β), vascular endothelial growth factor (VEGF), interferon-gamma-inducible Protein 10 (IP-10), and monocyte chemoattractant protein-1 (MCP-1), are in higher levels in patients with osteoarthritis [16]. Clinicians have to perform differential diagnosis of osteoarthritis from other diseases with knee pain, such as inflammatory arthritis (i.e., rheumatoid and psoriatic arthritis) or infectious arthritis (i.e., gout arthritis). The incidence of osteoarthritis increases with age and is affected by gender, with women experiencing higher rates than men [17]. The prevalence of osteoarthritis varies in different countries and increases as age limits increase and life expectancy increases, resulting in higher rates in older people worldwide [18]. Other studies define the prevalence of osteoarthritis at 30–50% for people under 65 years, while 80% of the elderly develop osteoarthritis in at least one joint [19]. Regarding the knee, osteoarthritis is the most common condition occurring in the knee joint, and the incidence increases with age [20]. The prevalence of knee osteoarthritis ranges from 16% to 22.9% worldwide [21]. Furthermore, obese patients are in more danger of the onset of osteoarthritis compared to non-obese patients (19.7% vs. 10.9%, respectively) [22]. According to race, African American women have a higher prevalence of knee osteoarthritis compared to Caucasian women (51% vs. 46.8%, respectively), whereas African American men have also a higher prevalence than the Caucasian men (40.9% vs. 36.7%, respectively), but distinctly lower than the prevalence in women [23]. The reduction in lean muscle mass, a main feature of sarcopenia, plays an important role in the occurrence of osteoarthritis, and often, these two diseases coexist, although only a few studies have been conducted to show the parallel comorbidity and how they are interrelated [24]. With regard to knee osteoarthritis in particular, it appears that individuals with sarcopenia and obesity are more prone to it. [25]. The onset of osteoarthritis is manifested by pain in the affected joint and the progressive limitation of the individual’s movements and functionality. The definitive treatment of the disease is admission for surgery for joint replacement, and at the early stage, relief can be provided by weight loss and exercise [26]. The loss of functionality of the individual and the surgery, with rehabilitation required, is a costly procedure, which places a significant burden on the health care system. Patients, in their lifespan, spend around USD 15,000 for medical costs [14]. On the other hand, sarcopenia has been shown to prolong hospitalization and rehabilitation time for hospitalized and surgical patients, thus increasing the cost of health care [27].

Total knee arthroplasty is a widely used operation and a life-saving procedure for many patients with chronic osteoarthritis, the rates of which are increasing as the average age of the population increases. As the rate of surgery increases and the years go by, the rates of revision total knee arthroplasty are also increasing, either due to the advanced age of the implements or due to an infection [28,29]. Infection of the operated joint or area and loosening of the joint are the most common complications, which can lead to a second surgery in a shorter than expected time frame. A total of 30% of cases that undergo revision have suffered an infection, which is the most common failure factor for both the first surgery and revision arthroplasty [30].

In view of these considerations, it is important to investigate the risk factors that slow down the patient’s recovery and prolong the hospitalization and/or the recovery time after total knee arthroplasty. Sarcopenia, since it often coexists with osteoarthritis and affects the patient’s general health status, should be diagnosed early and, if possible, be prevented before the onset of symptoms. However, there are limited available data about sarcopenia in patients who undergo total knee arthroplasty. In addition, the impact of sarcopenia coexisting with osteoarthritis on the time of recovery after surgery, as well as on other osteoarthritis symptoms, remains unclear. Thus, we aimed to investigate whether patients with coexisting sarcopenia and osteoarthritis had an improvement in osteoarthritis symptoms (i.e., pain, difficulty in daily activities, etc.) after total knee arthroplasty surgery compared to patients without sarcopenia, and the impact of sarcopenia on patients’ recovery after surgery.

## 2. Materials and Methods

### 2.1. Study Design and Participants

To conduct this study, 20 Greek subjects, (11 women and 9 men; mean difference (MD) of sarcopenic group: 64.8; MD of non-sarcopenic group: 76.4) who were admitted to Papageorgiou Hospital’s 2nd Orthopedic Department for total knee arthroplasty from November 2021 to April 2022, were enrolled and examined. All patients had been previously diagnosed with knee osteoarthritis and had undergone the necessary tests for its diagnosis and for planning the surgery. Patients excluded from the study were those who did not eventually undergo surgery due to concomitant health problems.

Out of all the patients enrolled, 20 patients with osteoarthritis who were to undergo total knee arthroplasty surgery were evaluated. These patients were divided into two categories based on the Foundation for the National Institutes of Health (FNIH) criteria, sarcopenic and non-sarcopenic [31]. Among the patients, there were 9 males and 11 females, with ages ranging from 49 to 84 years. Follow-up was performed after a period of rehabilitation of three months.

All patients gave written consent for their participation in this study. This study was conducted according to the Declaration of Helsinki, and it was approved by the Ethic Institutional Committee (28551/22 September 2022).

### 2.2. Anthropometry

The patients were weighed on a precision electronic balance (Seca 711) to the nearest 0.1 Kg, with light clothing and without footwear. Height was measured with a calibrated tape measure to the nearest 0.1 cm, without shoes. Waist and hip circumferences were also measured in the upright position with a calibrated tape measure. For waist circumference, the tape measure was placed at the narrowest point under the ribs, while for hip circumference, it was placed at the widest part of the buttocks. The measurements were taken at baseline, one day before the total knee arthroplasty.

### 2.3. Sarcopenia Measurement

The patients were divided into two categories, sarcopenic and non-sarcopenic, based on FNIH criteria [31]. We used both handgrip strength and lean mass indices, appendicular lean mass (ALM) and ALM adjusted to body mass index (ALM/BMI), to categorize patients. Based on FNIH criteria, cut-off values for grip strength were <26 kg and <16 kg for men and women, respectively. Cut-off values for ALM were <19.75 kg for men and <15.02 kg for women, and cut-off points for ALM/BMI were <0.789 and <0.512 for men and women, respectively [31].

Handgrip strength was measured by using an electronic hand dynamometer (Takei 5401). The Takei 5401 dynamometer has a digital display with a measuring range from 5 to 100 kg and can be used with both the left and right hands [32].

Body composition measurement was performed with a Bodystat Quadscan 4000 bioelectric impedance analyzer. BIA gives reliable measurements of extracellular fluid and can estimate a patient’s body fat and muscle mass [33]. Both ALM and ALM/BMI were used to diagnose and categorize patients. These indices showed us the amount of muscle mass. An algorithm based on height, resistance (R) to bioelectric impedance at 50 Hz, weight, and gender was used to calculate ALM in subjects up to 80 years of age.
ALM = 4.957 + (0.196 × height^2^/R) + (0.060 × weight) − (2.554 × gender)where for gender, men = 1, and women = 0 [34].

For patients older than 80 years, the algorithm is different.
ALM = 0.827 + (0.19 × impedance index) + (2.101 × gender) + (0.060 × weight)
where for gender, similarly, men = 1, and women = 0 [35].

All patients were measured with a digital handgrip dynamometer (Takei 5401), and body composition was measured by the BIA method (Quadscan 4000).

### 2.4. Knee Injury and Osteoarthritis Outcome Score (KOOS)

The KOOS is divided into five chapters/categories and includes a total of 42 questions. The 1st chapter refers to the pain experienced by the patient throughout daily movements and/or activities, the 2nd to accompanying symptoms, the 3rd to the difficulty the patient experiences in carrying out daily activities, the 4th to the patient’s functionality in sports activities, and the 5th to the reduction in quality of life due to limited mobility. To estimate the total score, each chapter is first calculated separately and then reduced to a percentage [36].

One day before surgery, the participants were asked to complete the KOOS to assess their knee condition [37].The KOOS score has been described as a reliable and valid criterion for assessing pain, as well as other symptoms, such as stiffness and functional limitations when performing daily activities, in patients with knee osteoarthritis [38]. The patients were asked to complete the questionnaire again three months after surgery, and the new score was compared with the first one in order to evaluate the improvement of each patient’s condition. The KOOS is a reliable criterion to evaluate a patient’s functional status and pain, both before, as mentioned above, and after surgery [39]; this way, we are able to estimate the patient’s rehabilitation progress and the success or not of total knee arthroplasty. An increase in the KOOS compared to pre-surgery levels implies an improvement in the patient’s condition. The results were pooled, and a comparison was made between the two groups of patients.

### 2.5. Statistical Analysis

Statistical analysis was performed using the statistical package SPSS v.27. Initially, a Shapiro–Wilk test was performed in order to examine the normal distribution of all dependent variables. It was found that all dependent variables except “Illness biomarker” (*p* < 0.001 for non-sarcopenic patients) and “Waist-to-Hip” (*p* = 0.005 for non-sarcopenic and *p* = 0.046 for sarcopenic patients) followed the normal distribution within each group (*p* > 0.05 for all cases). Based on this, the following statistical tests were performed. Student’s t-test was performed on independent samples to compare the mean values of all dependent variables between the two groups (non-sarcopenic and sarcopenic), except for “Disease Index” and “Waist/Speech Ratio”. The non-parametric Mann–Whitney test was used for the comparison between the two groups for the variables “Disease Index” and “Waist/heel ratio”. Two-way repeated measures analysis of variance (ANOVA) was performed to compare the mean values of the dependent variable KOOS between the two groups (non-sarcopenic and sarcopenic) before and after surgery (main effects and interaction), as previously described [40]. Results are presented as mean ± standard deviation, and the level of statistical significance was set at the *p* < 0.05 level.

## 3. Results

Of the total of 20 patients that were evaluated, 11 women and 9 men, 5 patients were found to belong to the sarcopenic group according to the FNIH criteria, and 15 to the control group. Three patients of the sarcopenic group were male, and two were female. For each group, the mean of anthropometric and metabolic variables was calculated, as presented in Table 1. We can see the comparison of the parameters between the two groups. The two groups showed a statistically significant difference in terms of age (*p* = 0.038), with the sarcopenic subjects being older than the non-sarcopenic subjects; however, no statistically significant differences were observed in terms of other body and metabolic characteristics (*p* > 0.01).

The patients’ muscle strength was calculated based on handgrip strength. The limits of muscle strength differed for the two genders, with 26 kg of handgrip strength defined as the minimum normal for men and 16 kg for women. No statistically significant difference was observed between sarcopenic (19.5 ± 8.2) and non-sarcopenic (25.2 ± 9.2) patients in terms of handgrip strength (*p* = 0.257). In contrast, a statistically significant difference was observed between sarcopenic and non-sarcopenic patients in lean mass, measured either as ALM (15.2 ± 4.00 vs. 20.0 ± 3.7, respectively (*p* = 0.023)) or as ALM/ΒΜΙ (0.50 ± 0.20 vs.0.70 ± 0.14, respectively; *p* = 0.007) (Figure 1a–c).

Two-way repeated measures ANOVA showed a statistically significant interaction (time x group) (*p* = 0.03) and a significant main effect of “time” (*p* < 0.01). In contrast, there was no statistically significant main effect of “group” (*p* = 0.74). Post-hoc analyses showed that the groups did not differ significantly before (0.38 ± 0.09 vs. 0.35 ± 0.09; *p* = 0.312) or after (0.54 ± 0.08 vs. 0.59 ± 0.10; *p* = 0.909) surgery. However, both sarcopenic (*p* = 0.06) and non-sarcopenic (*p* < 0.001) patients showed improvement after surgery on the KOOS index (Table 2 and Figure 2).

## 4. Discussion

The aim of this study was to investigate the improvement of osteoarthritis symptoms, such as pain, in patients with and without sarcopenia after arthroplasty surgery. In the present study, patients with confirmed osteoarthritis were examined and categorized into sarcopenic and non-sarcopenic groups based on muscle strength and lean mass. Muscle strength was calculated using a handheld dynamometer. Comparison of the results for handgrip strength showed that the mean values in the non-sarcopenic group were higher than in the sarcopenic group, but this difference between the two groups was not statistically significant. The lean mass of the patients was calculated based on the ALM index, according to the measurements made with the bioelectric impedance device. Lean mass showed a statistically significant difference between the sarcopenic patients and the non-sarcopenic group. The two groups showed no statistically significant difference in the KOOS, before and after surgery, but a significant difference showed after surgery in each group. Sarcopenic patients had a slightly higher mean score. The score of both groups increased significantly after surgery. The non-sarcopenic patients showed a greater improvement in score 3 months after; however, the effect of time seemed to be more significant than the effect of group.

There are not sufficient data available about sarcopenia in patients undergoing arthroplasty, although sarcopenia is associated with osteoarthritis, as shown by previous studies [32]. The prevalence of patients with sarcopenia awaiting for total knee arthroplasty is 35.5% in patients aged 56–87 years old; the prevalence of severe sarcopenia increases with age [6]. The presence of sarcopenia in patients with osteoarthritis could be secondary, as severe pain and poor physical function limit the ability of patients for activities, increasing the risk of its appearance [6]. A cohort study with patients >65 years old showed that patients with sarcopenia had worse outcomes after knee arthroplasty than the non-sarcopenic patients [41]. In contrast to this study, Ho et al. (2021) [42] found that both patients, aged >50 years old, with and without sarcopenia improved their functional capacity after total knee arthroplasty surgery. This result is in total agreement with our results. A meta-analysis to be carried out by Wang et al. (2022) [32] may provide a better explanation about the association between osteoarthritis and sarcopenia. Moreover, it is known that obesity and sarcopenic obesity are risk factors for the onset of osteoarthritis, especially for women [43]. Nevertheless, sarcopenia has, independently, a negative impact on the therapeutic effects of rehabilitation on physical mobility in patients undergoing total knee arthroplasty, as shown by a previous study [44]. On the other hand, the KOOS is a reliable questionnaire that assesses the patients’ condition before and after total knee arthroplasty surgery [44]. Notably, one year after surgery, the patients’ condition was significantly improved in terms of functionality and pain [44].

To the best of our knowledge, this was the first study that tried to investigate the impact of sarcopenia in patients with osteoarthritis on their rehabilitation progress, and osteoarthritis symptoms after total knee arthroplasty, compared to patients with osteoarthritis without sarcopenia. Regarding the limitations of this study, it is worth noting that age group plays an important role in both sarcopenia and osteoarthritis. The mean age of the two study groups, sarcopenic and non-sarcopenic, showed a statistically significant difference, with a higher mean age for the sarcopenic group. So, the two types of sarcopenia were probably mixed. However, in the non-sarcopenic group, there was only one patient who was 49 years old, which significantly lowered the mean. The handgrip strength measurement for the two groups showed a small difference, not significant in the mean measurements. The patients with sarcopenia had a lower mean handgrip strength. In addition, while both groups showed a significant increase in KOOS after surgery, the sarcopenic group showed a smaller mean increase in score. However, the difference between the two groups was not statistically significant, which may be due to the sample size. The study sample was limited, with only 20 patients, due to the reduced number of surgeries performed in the previous period. Finally, the KOOS is based on patients’ reports, which may be influenced to some extent by their current psychology and/or the different sense of pain that undoubtedly exists between people.

## 5. Conclusions

Both the sarcopenic patient group and the control group showed no significant differences between them in the scores for the assessment of the affected limb in either phase of the questionnaire, but a significant improvement was depicted in each group after arthroplasty surgery. The sarcopenic patients had a smaller increase in KOOS, but no statistically significant difference. Although there was a statistically significant group*time interaction, it appeared that there was no significant difference between the groups. Further investigation is strongly recommended with a larger sample and longer recovery time in order for more precise conclusions to be drawn and confirm the present results.

## Figures and Tables

**Figure 1 diseases-11-00036-f001:**
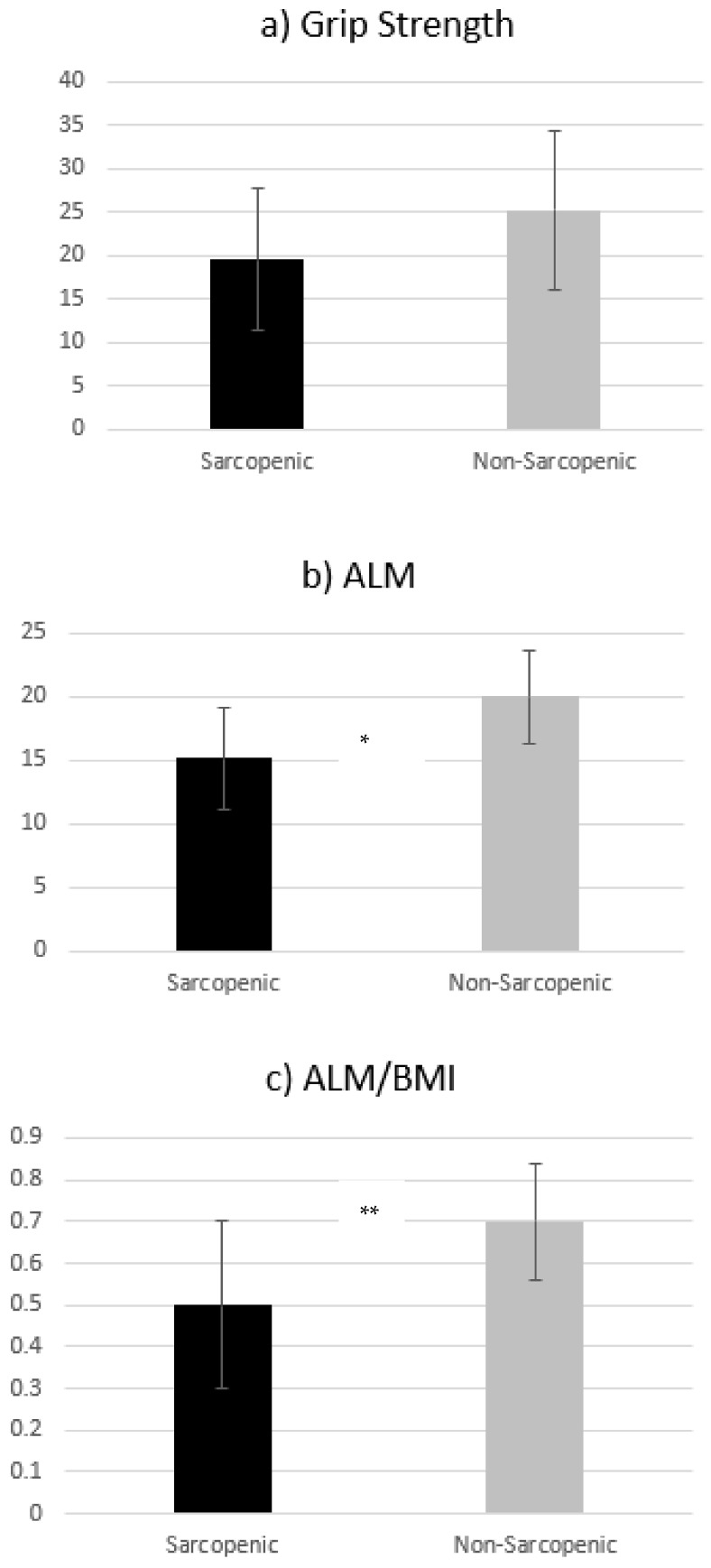
Differences between sarcopenic and non-sarcopenic patients, as categorized according to Foundation for the National Institutes of Health (FNIH) criteria. (**a**) Grip strength of sarcopenic and non-sarcopenic patients; (**b**) ALM of sarcopenic and non-sarcopenic patients; (**c**) ALM/BMI of sarcopenic and non-sarcopenic patients. ALM: appendicular lean mass; ALM/BMI: appendicular lean mass adjusted for body mass index; * *p* < 0.05; ** *p* < 0.01.

**Figure 2 diseases-11-00036-f002:**
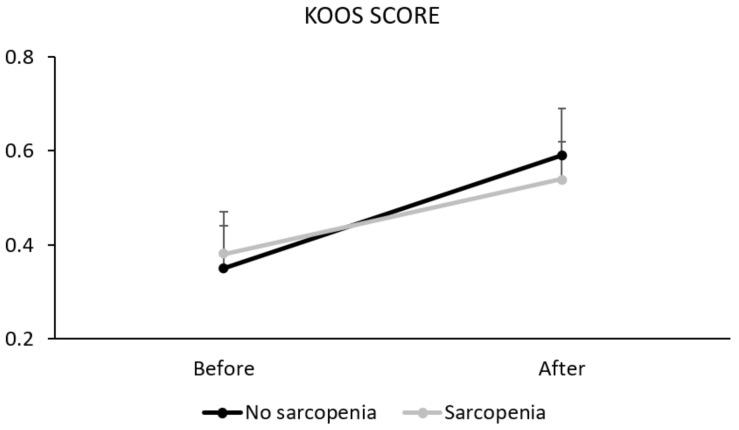
Knee Injury and Osteoarthritis Outcome Score (KOOS) before and after arthroplasty for sarcopenic and non-sarcopenic patients.

**Table 1 diseases-11-00036-t001:** Characteristics of sarcopenic vs. non-sarcopenic patients and differences between two groups.

Characteristic	Non-Sarcopenic (n = 15)	Sarcopenic (n = 5)	*p*-Value
Age (years)	64.8 ± 10.51	76.4 ± 8.05	0.038 *
Body Mass Index	29.39 ± 4.18	29.54 ± 6.20	0.953
Body Weight (kg)	84.14 ± 18.84	83.26 ± 14.14	0.925
Adipose Tissue (%)	37.55 ± 11.49	31.30 ± 5.47	0.125
Adipose Tissue (kg)	31.31 ± 10.48	25.78 ± 6.96	0.290
Fat Mass (kg)	52.77 ± 16.50	53.88 ± 8.30	0.888
Resting Metabolic Rate (kcal)	1618 ± 414	1655 ± 201	0.853
Waist/Hip Ratio	0.89 ± 0.18	0.71 ± 0.30	0.553
Nutritional Index	0.45 ± 0.02	0.45 ± 0.03	0.529
Disease Index	0.80 ± 0.08	0.84 ± 0.01	0.197

* *p*-value < 0.05 was considered to be statistically different. Values are presented as means ± standard deviations.

**Table 2 diseases-11-00036-t002:** Two-way ANOVA for repeated measures assessing the significance of time and sarcopenic and non-sarcopenic groups at KOOS before and after arthroplasty.

Parameters	Non-Sarcopenic		Sarcopenic		*p*-Value		
KOOS	Before	After	Before	After	T	G	T * G
	0.35 ± 0.09	0.59 ± 0.10 *	0.38 ± 0.09	0.54 ± 0.08 *	<0.01	0.74	0.03

(*) statistically significant difference compared to “before” values within the same group (*p* < 0.01); T: time; G: group.

## Data Availability

The data presented in this study are available upon request from the corresponding author.

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
