# Peer review of "Comparative Analysis on the Effect of Sarcopenia in Patients with Knee Osteoarthritis before and after Total Knee Arthroplasty"

_diseases, 2023, doi:10.3390/diseases11010036_

Round 1

Reviewer 1 Report

This is an article on the effect evaluation of sarcopenia and normal group before and after total knee replacement. I think have the following problem:

The first is the small number of cases, including 5 cases of sarcopenia and 15 cases of normal group.

The second is that there is a statistically significant age difference in demographic characteristics between the two groups.

The third problem is that the preface of the article points out that sarcopenia is a kind of decreased defense ability of the immune system, which leads to decreased response ability to infection, but the result does not mention this aspect, the preface research question is not very good, and the conclusion of the article cannot be related to the hypothesis.

The fourth question is that the research purpose of this paper is to explore the relationship between sarcopenia and osteoarthritis, because sarcopenia can lead to osteoarthritis, and osteoarthritis can also cause muscle atrophy due to pain, so it is not clear that muscle atrophy is caused by sarcopenia.

Author Response

Thank you very much for your comments. They were very helpful in improving our manuscript. We took them into consideration and replied to them point by point.

Reviewers’ comments

Reviewer 1

R: The first is the small number of cases, including 5 cases of sarcopenia and 15 cases of normal group. The second is that there is a statistically significant age difference in demographic characteristics between the two groups.

A: Indeed, the sample size is small and there is a statistically significant difference in the age between the two groups. These limitations of the study are referred and discussed in the manuscript. In general, it is difficult to collect a larger sample size in a reasonable time with this disease features. However, it is in our future plans to extend our study in larger sample size, which will also allow us to have age-matched group.

R: The third problem is that the preface of the article points out that sarcopenia is a kind of decreased defense ability of the immune system, which leads to decreased response ability to infection, but the result does not mention this aspect, the preface research question is not very good, and the conclusion of the article cannot be related to the hypothesis.

A: We don't examine the inflammation or the immune system of participants. Thus, we deleted from the introduction the mention about the immune system. Furthermore, we already change the aim of the study.

R: The fourth question is that the research purpose of this paper is to explore the relationship between sarcopenia and osteoarthritis, because sarcopenia can lead to osteoarthritis, and osteoarthritis can also cause muscle atrophy due to pain, so it is not clear that muscle atrophy is caused by sarcopenia.

A: The purpose of our study was not stated clear. The purpose was to investigate the role of sarcopenia on the recovery of patients who underwent to knee arthroplasty surgery. We now revised the phrase referring the aim of our study

Reviewer 2 Report

The manuscript should be withdrawn, rephrased according to the new goal being in line with hypothesis and once again submitted.

For the first look, the study seems to be adequate, but during reading many doubts rise. The two types of sarcopenia are mixed. Authors wanted to clarify the relation between OA and sarcopenia, and its progress after surgery, but it is not possible to use the two groups of patients among which one group is consisted of elders. In this case it is very hard to show the goal. The elder patients subjected to the sarcopenic group suffer from the primary sarcopenia so called age-related sarcopenia. the oldest patient was 84 years old, while in the non-sarcopenic group the youngest was 49. If primary age-related sarcopenia start to occur about 50 years, and accelerate after 60, it is expected that this patient is not sarcopenic (and this patient was included into no-sarcopenic group). But, reading the title, it is expected that the real relationship between OA and sarcopenia will be presented. That means the patients without old-related sarcopenia and suffering from knee osteoarthritis will be studied. It would to allow to present real relationship between OA and sarcopenia disease-related so called secondary sarcopenia. Such patients suffering from the pain suffer also from immobilization which influence disease-related sarcopenia. The pain disappears after surgery, and patients can became more active, and prevent acceleration of sarcopenia, especially as they get older.

In conclusion, the goal of the study should be changed before publication, because in this for the paper is not in line with hypothesis.

1. Introduction of Abstract is not understood. There are the two types of sarcopenia: primary and secondary. There is not “ possibility of appearance” but it occurs if we want or not.

2. Abstract, Purpose – rather knee OA is responsible for the progress of sarcopenia due to the immobilization

3. Introduction – it should be clarify what type of sarcopenia is discussed. Primary or secondary

4. OA results from the reduction of lean muscle mass? Add the reference, please

5. If sarcopenic obesity is mentioned, there should be the difference between obesity and sarcopenic obesity given

6. “Therefore, there are limited available data about sarcopenia in patients who undergo in arthroplasty.” – but there should be clarified at what age the patients were considered

Author Response

Thank you very much for your pertinent comments. They were very helpful in improving our manuscript. We have taken them into consideration and have replied to them point by point. 

Reviewer 2

R: For the first look, the study seems to be adequate, but during reading many doubts rise. The two types of sarcopenia are mixed. Authors wanted to clarify the relation between OA and sarcopenia, and its progress after surgery, but it is not possible to use the two groups of patients among which one group is consisted of elders. In this case it is very hard to show the goal. The elder patients subjected to the sarcopenic group suffer from the primary sarcopenia so called age-related sarcopenia. the oldest patient was 84 years old, while in the non-sarcopenic group the youngest was 49. If primary age-related sarcopenia start to occur about 50 years, and accelerate after 60, it is expected that this patient is not sarcopenic (and this patient was included into no-sarcopenic group). But, reading the title, it is expected that the real relationship between OA and sarcopenia will be presented. That means the patients without old-related sarcopenia and suffering from knee osteoarthritis will be studied. It would to allow to present real relationship between OA and sarcopenia disease-related so called secondary sarcopenia. Such patients suffering from the pain suffer also from immobilization which influence disease-related sarcopenia. The pain disappears after surgery, and patients can became more active, and prevent acceleration of sarcopenia, especially as they get older.

In conclusion, the goal of the study should be changed before publication, because in this for the paper is not in line with hypothesis.

A: We re-write the purpose of the manuscript. The purpose was to investigate the role of sarcopenia on the recovery of patients who underwent to knee arthroplasty surgery. It is also in our future plans to extend our study in larger sample size, which will also allow us to have age-matched group.

R: 1. Introduction of Abstract is not understood. There are the two types of sarcopenia: primary and secondary. There is not “ possibility of appearance” but it occurs if we want or not.

A: We revise the introduction of the abstract.

R: 2. Abstract, Purpose – rather knee OA is responsible for the progress of sarcopenia due to the immobilization

A: We re-write out purpose such as wan comprehensive. We are not investigating the progress of sarcopenia, but the recovery of patients who underwent in arthroplasty.  

R: 3. Introduction – it should be clarify what type of sarcopenia is discussed. Primary or secondary

A: We do not investigate the type of sarcopenia. We just investigate if sarcopenic patients has worse or better recovery after the surgery in the contrary of non-sarcopenic patients.

R: 4. OA results from the reduction of lean muscle mass? Add the reference, please

A: Reference has been added.

R: 5. If sarcopenic obesity is mentioned, there should be the difference between obesity and sarcopenic obesity given

A: We mention about the difference between obesity and sarcopenic obesity in the discussion.

R: 6. “Therefore, there are limited available data about sarcopenia in patients who undergo in arthroplasty.” – but there should be clarified at what age the patients were considered

A: In the discussion we mentioned relevant studies and also the age groups of the participants.

Reviewer 3 Report

I believe that this cross-sectional study is suitable for publication in Diseases MDPI. Please see below some suggestion:

a)       Additional works, including meta-analyses, evaluated the relationship between sarcopenia osteoarthritis. These references should be included and discussed in the discussion PMID: 35921336, PMID: 31089492 https://doi.org/10.1371/journal.pone.0272284

b)      Please revise the work for the presence of numerous typo errors

c)       The font style of the citations being included throughout the text is not uniformed with that of the main text

d)      Authors are encouraged to reduce the length of the abstract. This will improve its readability

e)      Please include patients ages and gender ration in the methods

f)        Please include supporting references in the statistics. For the ANOVA test, authors are kindly encouraged to include this reference in which the method is extensively applied (DOI: 10.3389/fmicb.2021.789991)

g)       Yellow annotation in all figures should be removed.

h)      Please uniform the style of the graphs in figure 2)

g)       Yellow annotation in all figures should be removed.

h)      Please uniform the style of the graphs in figure 2

Author Response

Thank you very much for your comments. We made all the changes you 've proposed in order to improve our manuscript. 

Reviewer 3

R: a)       Additional works, including meta-analyses, evaluated the relationship between sarcopenia osteoarthritis. These references should be included and discussed in the discussion PMID: 35921336, PMID: 31089492 https://doi.org/10.1371/journal.pone.0272284

Α: We discussed about those references in the discussion

R: b)      Please revise the work for the presence of numerous typo errors

Α: Typo errors checked and corrected

R: c)       The font style of the citations being included throughout the text is not uniformed with that of the main text

A: Reference style of the manuscript revised according to the journals’ guidelines.

R: d)      Authors are encouraged to reduce the length of the abstract. This will improve its readability

Α: We reduce the length of the abstract

R: e)      Please include patients ages and gender ration in the methods

Α: We include mean age, gender and ration in the methods.

R: f)        Please include supporting references in the statistics. For the ANOVA test, authors are kindly encouraged to include this reference in which the method is extensively applied (DOI: 10.3389/fmicb.2021.789991)

Α: We add the reference for ANOVA

R: g)       Yellow annotation in all figures should be removed.

Α: Yellow annotation removed from the figures

R: h)      Please uniform the style of the graphs in figure 2)

Α: We changed the format of the graphs to make them uniform

Round 2

Reviewer 1 Report

The symptoms of osteoarthritis are basically improved after TKA. The purpose of your study is whether it is more appropriate to replace patients with sarcopenia combined with knee OA to improve the symptoms of sarcopenia after TKA?

Author Response

Thank you very much for your comments. 

The aim of the study was to investigate if patients with sarcopenia have better or worse rehabilitation on osteoarthritis outcomes, such as pain (which was examined by using KOO score), than the patients without sarcopenia after arthroplasty surgery.

We also noticed in the conclusion that the osteoarthritis symptoms improved after the arthroplasty.

Reviewer 2 Report

I do not have more comments, however the text should be corrected by native e.g. " Secondary sarcopenia related with the presence of a disease.". it seems that there should be "relates" or "is related". 

Author Response

Thank you very much for your comments.

Our manuscript has been checked and corrected by a native speaker.

Round 3

Reviewer 1 Report

The author should further clarify the purpose of the study. My understanding is whether the sarcopenia combined with osteoarthritis group and the osteoarthritis alone group can benefit the treatment of sarcopenia after TKA.

Author Response

We re-write the aim of our study, and in the introduction, we try to explain more the rationale underlying our purpose.